# Pretraining A Large Language Model using Distributed GPUs: A Memory-Efficient Decentralized Paradigm

## Abstract

Pretraining large language models (LLMs) typically relies on centralized clusters equipped with hundreds or thousands of high-memory GPUs (*e.g.*, H100/A100), creating obstacles for a wide range of exploration in the community. Recent decentralized training methods reduce communication overhead by employing federated optimization; however, these methods still need to store and train the entire model on each node, remaining constrained by GPU memory limitations. In this work, we propose **SP**arse **E**xpert **S**ynchronization (**SPES**), a memory-efficient decentralized framework for pretraining mixture-of-experts (MoE) LLMs. SPES trains only a small subset of experts on each node during training, substantially reducing the memory footprint per node. Each node updates its local experts and periodically synchronizes with other nodes, eliminating the need to transmit the full set of model parameters and enabling efficient knowledge sharing across the distributed network. To accelerate convergence, we introduce an expert-merging warm-up strategy. Experts exchange knowledge via model merging in the early training stages, promoting faster establishment of the foundational capabilities for each expert. With SPES, we train a 2B-parameter MoE LLM using 16 standalone 48GB GPUs (NVIDIA L40S) over internet connections, which achieves competitive performance with centrally trained LLMs under similar computational budgets. We further demonstrate the scalability of SPES by training a model up to 7B parameters with open-source data, matching prior centralized baselines. Our SPES pre-training paradigm can be extended to more low-end GPUs and train LLM of larger scales. Code and models will be released.

## 1 Introduction

Large language models (LLMs) (Achiam et al., 2023; Grattafiori et al., 2024; Yang et al., 2025; Liu et al., 2024; Muennighoff et al., 2024) have shown strong generalization capabilities across various downstream tasks, establishing themselves as fundamental components in real-world applications such as conversational assistant (Cui et al., 2024) and embodied agent (Fung et al., 2025). However, pretraining LLMs remains highly resource-intensive. The main bottlenecks arise from the substantial GPU memory required to store model parameters, activations, optimizer states, and gradients, and the need of low-latency, high-bandwidth inter-device communication to support model and data parallelism (Shoeybi et al., 2019; Rasley et al., 2020; Zhao et al., 2023). Consequently, existing LLMs are typically trained under centralized settings (as shown in Fig. 1 (left)), utilizing co-located clusters equipped with high-memory GPUs and fast interconnects (*e.g.*, RDMA). For instance, LLaMA3-405B (Grattafiori et al., 2024) is trained using up to 16K H100 GPUs linked with high-bandwidth interconnects, while OLMo2 7B (OLMo et al., 2024) is trained on a cluster of 1,024 H100 GPUs. Such high infrastructure requirements make LLM pretraining inaccessible to most researchers in the community.

To mitigate the demands of centralized LLM training, recent works such as DiLiCo (Douillard et al., 2023) and Photon (Sani et al., 2024) have explored decentralized pre-training paradigms (as shown in Fig. 1 (middle)). In these approaches, each workstation performs local updates and synchronizes with peers intermittently via a parameter server, following a federated optimization protocol (*e.g.*, FedAvg (McMahan et al., 2017)). This sparse communication mode significantly reduces the

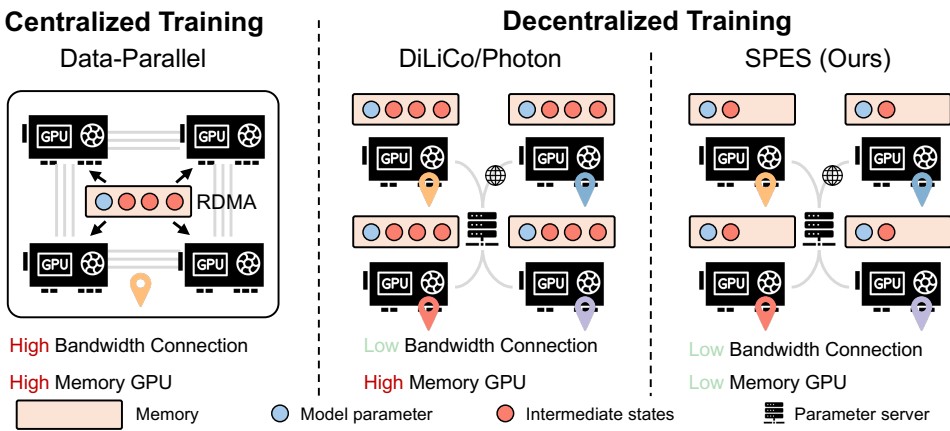

Figure 1: **Comparison of different pretraining paradigms for LLM. Left**: centralized training, which requires high-memory GPUs and high-bandwidth interconnects (*e.g.*, RDMA) for its tightly coupled model or data parallelism. **Middle**: existing decentralized training (*e.g.*, DiLiCo, Photon), where each node trains a full model locally, reducing bandwidth needs but still demanding high-memory GPUs. **Right**: our proposed SPES, a memory-efficient decentralized method for training MoE-based LLMs, where each node trains only a subset of experts, substantially reducing both per-GPU memory usage and communication overhead.

bandwidth requirements compared to centralized data- or model-parallel methods, enabling training across geographically distributed, heterogeneous GPU clusters. While communication constraints are relaxed, however, these approaches still require each node to update the full set of model parameters. Consequently, the memory footprint per node remains substantial. This limitation is especially significant for training large-scale LLMs, where insufficient memory can be a bottleneck.

To address this challenge, we propose **SP**arse **E**xpert **S**ynchronization (SPES), a memory-efficient, decentralized training paradigm tailored for MoE-based LLMs, as illustrated in the right panel of Fig. 1. Compared to dense models, MoE models are inherently well-suited for decentralized environments, as each expert can be managed independently, enabling finer-grained training and resource management. In SPES, each node is responsible for training a distinct subset of experts, while keeping the remaining experts frozen during local updates. This design substantially reduces the memory requirement per node, since each node only needs to maintain the gradients and optimizer states for the experts assigned to it [1]. All nodes periodically synchronize their trained experts with peers, ensuring continuous knowledge sharing across the network. By eliminating the need to transmit the entire model weights, this sparse synchronization approach substantially reduces communication overhead and enables efficient knowledge exchange between nodes. A challenge in this sparse training regime is the limited token utilization of individual experts, as each expert is trained on only a subset of the total training tokens, which can slow down model convergence. To address this issue, we introduce an expert-merging warm-up strategy: in the early stages of training, we periodically merge each expert with its most similar peers in a weighted average manner, accelerating the knowledge acquisition of each expert.

We evaluate the effectiveness of SPES by pretraining MoE LLMs on both 2B and 7B scales in decentralized settings. Our results show that SPES enables the training of a 2B-parameter MoE LLM on 16 standalone NVIDIA L40S GPUs (48GB) over internet, achieving performance comparable to centrally trained models under comparable computational budgets. Compared with previous decentralized training frameworks, SPES reduces up to 33.3% communication cost and significantly lowers per-GPU memory requirements. We further demonstrate the scalability of SPES by training a 7B MoE LLM with open-source datasets, achieving performance on par with previous models trained with similar data and compute resources. Ablation studies and in-depth analysis are also provided to validate the design choices of SPES.

---

[1]Note that optimizer states and gradients typically dominate the static memory footprint (excluding activations) in model training. For example, AdamW (Loshchilov & Hutter, 2017) can consume up to 75% of the total static memory usage.

Our contributions can be summarized as follows. **(i) A memory-efficient decentralized pretraining framework.** We propose SPES, a memory-efficient decentralized framework for pretraining large MoE-based LLMs, where each node trains only a subset of experts, significantly reducing per-device memory and communication overhead. **(ii) An expert-merging warm-up strategy.** We introduce an expert-merging warm-up strategy to periodically aggregate similar experts during early training, enabling stronger expert representations with sparse decentralized training. **(iii) Empirical results.** We demonstrate the effectiveness of SPES by training 2B and 7B MoE LLMs using publicly available datasets on weakly connected GPUs. SPES achieves competitive performance but with significantly lower communication and memory costs compared to previous approaches.

As most existing decentralized LLM training frameworks are not open-sourced, we implement a custom server-client communication protocol based on gRPC (gRPC, 2015) and integrate it into a mainstream LLM pretraining framework (Muennighoff et al., 2024). Our model and code will be released to facilitate future research on decentralized training.

## 2 RELATED WORK

**Decentralized Training.** Decentralized training has been studied for both fine-tuning (Wu et al., 2025; Bai et al., 2024; Sun et al., 2024) and pretraining (Douillard et al., 2023; Sani et al., 2024; Jaghouar et al., 2024) LLMs. The works on finetuning pretrained LLMs usually target for privacy-preserving adaptation. FATE-LLM (Fan et al., 2023) explores federated fine-tuning for advertising generation. Subsequent works (Kuang et al., 2024; Zhang et al., 2024a; Ye et al., 2024) extend federated LLM fine-tuning to instruction-tuning settings. To reduce communication and memory costs, parameter-efficient federated fine-tuning methods have been proposed, such as FedLoRA (Yi et al., 2023) and FedPETuning (Zhang et al., 2023). The works on pretraining LLMs train LLMs from scratch under communication constraints. DiLiCo (Douillard et al., 2023) and Photon (Sani et al., 2024) are among the first to study decentralized LLM pretraining. With FedAvg (McMahan et al., 2017), they achieve comparable perplexities to centrally trained models while substantially reducing communication cost. Jaghouar et al. (2024) introduced INTELLECT-1, a 10B-parameter LLM pretrained across multiple independent computing devices, and Charles et al. (2025) demonstrated the scalability of this communication-efficient paradigm. Despite such advances, those methods still incur significant memory and communication overhead due to full-model training and synchronization. In contrast, our SPES only needs to train a subset of parameters per node, substantially reducing both the memory and communication costs.

**Memory-Efficient Pretraining.** Methods to reduce memory in LLM pretraining primarily leverage sharding and parallelism on tightly coupled accelerators. Data parallelism such as ZeRO (Rajbhandari et al., 2020) and FSDP (Zhao et al., 2023) partition optimizer states, gradients, and model parameters, enabling distributed storage and computation. Model-parallel techniques (Shoeybi et al., 2019)—including pipeline, tensor, and expert parallelism—split model computation to accommodate larger architectures. However, these strategies typically assume a centralized cluster with high-bandwidth interconnects to facilitate frequent synchronization. Orthogonal techniques include mixed precision (Micikevicius et al., 2017), activation checkpointing, memory-efficient attention (Dao et al., 2022; Dao, 2023), and optimizer quantization (Dettmers et al., 2021). Our proposed SPES enables cross-node expert sharding with sparse synchronization: gradients and optimizer states are distributed across geographically heterogeneous nodes, each of which trains only the MoE experts assigned to it and communicates only necessary updates. SPES is designed for environments with heterogeneous, low-bandwidth interconnects, such as single-GPU nodes where intra-node sharding is infeasible. Moreover, SPES complements existing parallelism paradigms: when multiple GPUs are available per node, SPES can be combined with previous parallelism strategies to maximize memory efficiency and scalability.

## 3 MEMORY-EFFICIENT DECENTRALIZED PRETRAINING

In this section, we present the details of our proposed **SP**arse **E**xpert **S**ynchronization (**SPES**), a memory-efficient decentralized pretraining framework for MoE-based LLMs. SPES partitions expert training across weakly connected nodes and synchronizes weights intermittently, substantially reducing both the memory usage and the communication overhead compared to prior paradigms. We

begin with the preliminaries (Section 3.1), followed by the overview of the framework (Section 3.2), and the details of the SPES methodology (Section 3.3).

## 3.1 PRELIMINARIES

**Decentralized Training.** Let $\mathcal{S} = \{\eta_1, \ldots, \eta_N\}$ denote $N$ distributed nodes, each node $\eta_i$ having its local training data $\mathcal{D}_i$. Existing decentralized training frameworks such as DiLiCo (Douillard et al., 2023) decompose model optimization into two levels: an outer optimizer that governs global synchronization and an inner optimizer that performs local node updates. In the $t^{th}$ communication round, the global parameters obtained in the previous round, denoted by $\boldsymbol{\theta}^{(t-1)}$, are broadcast to all nodes. Each node runs $H$ steps the inner optimizer (*e.g.*, AdamW (Loshchilov & Hutter, 2017)) on its shard $\mathcal{D}_i$, producing the updated local parameters $\boldsymbol{\theta}_i^{(t)}$. The parameter server then aggregates the local updates by averaging the differences between the local and global models, and applies the outer optimizer to update the global parameters as follows:

$$\boldsymbol{\theta}^{(t)} \leftarrow \text{OuterOpt}\left(\boldsymbol{\theta}^{(t-1)}, \frac{1}{N}\sum\nolimits_{i=1}^{N}\left(\boldsymbol{\theta}_i^{(t)} - \boldsymbol{\theta}^{(t-1)}\right)\right). \tag{1}$$

When the outer optimizer is set to SGD, the above training procedures become the FedAvg (McMahan et al., 2017), which enables distributed training while minimizing communication overhead. However, each node is required to train the entire model, which needs to store a large amount of intermediate optimizer states, limiting its applicability to memory-constrained devices.

**Mixture-of-Experts (MoE) LLM.** MoE architectures (Lepikhin et al., 2020; Muennighoff et al., 2024; Dai et al., 2024) extend standard transformer-based LLMs by introducing a set of $M$ expert sub-networks $\{\mathcal{E}_j\}_{j=1}^{M}$, each sub-network $\mathcal{E}_j$ being parameterized by $\phi_j$. Given an input token $x$, a gating function $\mathcal{G}(x)$ is used to select a sparse subset of experts to process it. The output of the MoE block is computed as a weighted sum of the selected experts:

$$\text{MoE}(x) = \sum_{j \in \{1, \ldots, M\}} \mathcal{G}_j(x) \cdot \mathcal{E}_j(x), \tag{2}$$

where $\mathcal{G}_j(x)$ is the gating weight for the $j$-th expert. MoE enables scaling by activating only a subset of experts per token, thus increasing the model capacity without increasing the computation.

## 3.2 OVERALL FRAMEWORK

Previous sharding strategies, such as FSDP (Zhao et al., 2023) and ZeRO (Rajbhandari et al., 2020), partition LLM model training in centralized data-parallel setups. Each node is responsible for a subset of model modules, which alleviates individual memory constraints. However, when inter-node communication bandwidth is limited, the tight coupling between model shards may lead to suboptimal performance due to insufficient synchronization of model updates. To address this issue, we adopt the MoE architecture to train the LLM, where expert modules can be managed independently, thus relaxing synchronization requirements and enabling fine-grained resource allocation. Following prior works (Touvron et al., 2023; OLMo et al., 2024; Bai et al., 2023), We employ a standard decoder-only MoE LLM, which is composed of self-attention layers, sparse expert feed-forward networks selected via a routing mechanism, and normalization layers, as illustrated in Fig. 2(a). Positional encoding is implemented using RoPE (Su et al., 2024), SwiGLU (Shazeer, 2020) is adopted as the activation function, and normalization is performed with RMSNorm (Zhang & Sennrich, 2019). Bias terms are omitted to enhance stability. Specifically, we utilize the drop-less MoE (Gale et al., 2023), as suggested by Muennighoff et al. (2024), to maximize expert utilization.

In this work, our goal is to train an MoE-based LLM using distributed GPUs. Compared to traditional centralized training, the key challenge of our decentralized training lies in the memory and communication bottlenecks. We therefore propose Sparse Expert Synchronization (SPES) to solve this issue. As illustrated in Fig. 2(b), we take advantage of the inherent modularity of MoE LLM by distributing expert training across the $N$ nodes. Each node is assigned with some shared modules and a unique subset of the $M$ experts, allowing memory-efficient local updates. During training, the nodes perform efficient synchronization to share knowledge. To improve data utilization for each expert, we further propose an expert-merging warm-up strategy. The details of our SPES are presented in the following section.

Figure 2: (a) **Illustration of our model structure**, in which we utilize an MoE LLM comprising standard self-attention blocks, normalization layers, and routed feed-forward modules. (b) **Illustration of SPES**, where each node performs local training on a disjoint subset of experts to reduce memory consumption. During weight synchronization, only the trained parameters are transmitted to the parameter server, minimizing communication overhead. To improve data utilization, we propose an expert-merging strategy that merges similar experts to facilitate knowledge sharing.

### 3.3 SPARSE EXPERT SYNCHRONIZATION

**Expert Assignment and Local Training.** We denote by $\boldsymbol{\Phi} = \{\boldsymbol{\phi}_j\}_{j=1}^M$ the set of parameters of all experts. Refer to Fig. 2(b), we partition $\boldsymbol{\Phi}$ into $N$ disjoint subsets, so that $\boldsymbol{\Phi} = \boldsymbol{\Phi}_1 \cup \boldsymbol{\Phi}_2 \cup \ldots \cup \boldsymbol{\Phi}_N$, where $\boldsymbol{\Phi}_i$ denotes the subset of experts assigned to node $\eta_i$. We denote by $\overline{\boldsymbol{\Phi}}_i$ the set of unassigned experts for node $\eta_i$, and denote by $\boldsymbol{\psi}_i$ the parameters of the shared modules. At the start of each local training round $t$, node $\eta_i$ receives the global model parameters updated at round $t-1$ from the server and then performs $H$ rounds of local updates on its local data $\mathcal{D}_i$. The designated expert parameters $\boldsymbol{\Phi}_i$ and the shared parameters $\boldsymbol{\psi}_i$ will be optimized while keeping $\overline{\boldsymbol{\Phi}}_i$ fixed. The updated local parameters at round $t$ can be denoted as:

$$\boldsymbol{\theta}_i^{(t)} = \left( \boldsymbol{\psi}_i^{(t)}, \, \boldsymbol{\Phi}_i^{(t)}, \, \overline{\boldsymbol{\Phi}}_i^{(t-1)} \right). \tag{3}$$

In the above process, although each node stores a full copy of the model parameters, the gradients and optimizer states are only required exclusively for the parameters to be updated, thereby substantially reducing the per-node memory overhead.

**Sparse Synchronization.** At the end of each local training round $t$, node $\eta_i$ holds updated local parameters $\boldsymbol{\theta}_i^{(t)}$, where the shared parameter $\boldsymbol{\psi}_i$ and the assigned experts $\boldsymbol{\Phi}_i$ are updated. During synchronization, each node transmits the updated parameters to the server. Shared parameters are aggregated using FedAvg (McMahan et al., 2017), while experts are updated via direct assignment:

$$\boldsymbol{\theta}^{(t)} = \left( \frac{1}{N} \sum_{i=1}^N \boldsymbol{\psi}_i^{(t)}, \, \bigcup_{i=1}^N \boldsymbol{\Phi}_i^{(t)} \right). \tag{4}$$

The aggregated global parameters $\boldsymbol{\theta}^{(t)}$ are then broadcast to all nodes for the next round of training. By synchronizing only assigned experts and shared parameters, SPES substantially reduces communication overhead, enabling scalable decentralized training under limited bandwidth.

**Expert-Merging Warm-Up.** While achieving notable memory efficiency, SPES faces a practical challenge in sparse training: each node updates only its local experts, leaving many tokens assigned to frozen (unassigned) experts without contributing to gradient updates. This leads to lower token utilization compared to centralized training with an equivalent token budget. To address this issue, we propose an expert-merging warm-up strategy to improve token utilization. The core idea is to periodically merge parameters of similar experts across nodes during synchronization.

Instead of updating each expert solely with local assignments, we identify peer experts with similar input projections and merge their parameters to facilitate knowledge sharing. Specifically, for the $j$-th expert, we compute pairwise cosine similarities between input projection layers:

$$A_{j,k} = \frac{\langle \boldsymbol{w}_j^{\mathrm{in}}, \, \boldsymbol{w}_k^{\mathrm{in}} \rangle}{\|\boldsymbol{w}_j^{\mathrm{in}}\|_2 \, \|\boldsymbol{w}_k^{\mathrm{in}}\|_2}, \quad j, k \in \{1, \ldots, M\}, \tag{5}$$

where $\boldsymbol{w}_j^{\text{in}}$ denotes the input projection weights of the expert $\mathcal{E}_j$, for which we select the $K$ most similar experts $\mathcal{Q}_j = \text{TopK}_k(A_{j,k})$, excluding itself. The merged parameters for $\mathcal{E}_j$ are then computed using task arithmetic (Ilharco et al., 2022):

$$\widetilde{\boldsymbol{\phi}}_j^{(t)} = \boldsymbol{\phi}_j^{(t)} + \alpha \frac{1}{K} \sum_{k \in \mathcal{Q}_j} \left( \boldsymbol{\phi}_k^{(t)} - \boldsymbol{\phi}_j^{(t)} \right), \tag{6}$$

where $\alpha$ is a scaling factor. To preserve the specialization of experts in later training stages, we perform merging only in the initial $T_{\text{merge}}$ steps and linearly decay $\alpha$ to zero. This expert-merging strategy enables each expert to benefit from gradients from multiple nodes, which improves token utilization and accelerates knowledge acquisition in decentralized sparse training settings.

**Efficiency Analysis.** SPES achieves substantial improvements in both memory and communication efficiency compared to conventional decentralized training methods. For example, when using the AdamW optimizer, DiLiCo (Douillard et al., 2023) requires each node to store optimizer states and gradients for all model parameters, resulting in a memory cost of $4 \times (|\boldsymbol{\psi}| + |\boldsymbol{\Phi}|)$ and a communication cost of $2 \times N \times (|\boldsymbol{\psi}| + |\boldsymbol{\Phi}|)$ per round. In contrast, SPES exploits expert partitioning, and each node only needs to store the intermediate states for the shared parameters and the assigned experts, which reduces the per-node memory cost to $4 \times |\boldsymbol{\psi}_g| + |\boldsymbol{\Phi}| + 3 \times |\boldsymbol{\Phi}_i|$. Similarly, communication overhead is also significantly reduced, as only shared parameters and updated experts are synchronized, resulting in a cost of $N \times (2 \times |\boldsymbol{\psi}_g| + |\boldsymbol{\Phi}| + |\boldsymbol{\Phi}_i|)$ per round. SPES achieves significant reductions in both memory and communication cost, especially as the number of nodes increases. For instance, when training a 2B-parameter MoE model with 16 experts in 16 nodes (one GPU per node; see Fig. 3 for details), DiLiCo requires 55GB of memory per node, whereas SPES reduces this requirement to 35GB. In addition, SPES achieves a 33.3% reduction in communication cost.

**Training Losses.** Our model is trained with three losses: standard cross-entropy loss for next token prediction, z-loss (Chowdhery et al., 2023; Zoph et al., 2022) for enhancing training stability, and a load-balancing loss (Lepikhin et al., 2020) to encourage uniform expert utilization. Within each node, PyTorch FSDP and mixed-precision are used to further improve memory efficiency. For cross-node synchronization, we use our customized gRPC-based communication protocol.

## 4 EXPERIMENTS

### 4.1 EXPERIMENTS SETUP

**Implementation Details.** We conduct experiments by training our SPES models at three scales: 1B, 2B, and 7B parameters (see Table 1 for detailed configurations). All ablation studies are performed on the 1B model, while the 2B and 7B models are trained to compare with previous work. For the 7B model, our training is distributed over $N = 4$ compute nodes, each equipped with 8 NVIDIA A800 GPUs interconnected via NVLink. A parameter server with a 96-core Intel Xeon processor (2.90 GHz) and 1.44TB RAM is used for parameter aggregation. The nodes communicate with the server over a 13 Gbps Ethernet network, with each node training eight experts (approximately 2.5B trainable parameters per node).

For the 2B model, training is performed on $N = 16$ nodes, each hosting one NVIDIA L40S GPU. The parameter server comprises a 64-core Intel Xeon Gold 6148 (2.40 GHz) and 720GB RAM, with nodes connected via 17 Gbps Ethernet. Each node manages the training of one expert, resulting in roughly 0.7B trainable parameters per node. The expert merging warmup steps, $T_{\text{merge}}$, is set to 12,500 training steps, with merging executed for every 500 steps. The initial value of $\alpha$ is set to 0.1. All models are trained with AdamW optimizer (Loshchilov & Hutter, 2017). Additional implementation details are provided in the **Appendix A**.

**Training Data.** We train our models exclusively on publicly available datasets, ensuring accessibility for the research community. The 2B and 7B models are trained on data sampled from Ultra-FineWeb (Wang et al., 2025) and SlimPajama (Soboleva et al., 2023), complemented by openweb-math, algebraic stack, pes2o, arxiv, and StarCoder drawn from olmo-mix-1124 (OLMo et al., 2024) to provide domain-specialized coverage in reasoning, scientific, and programming knowledge. The 1B model is trained solely on SlimPajama for a lightweight and efficient pretraining. For tokenization, we use the tokenizer trained by Bai et al. (2023), which offers efficient subword segmentation

Table 1: **Model configurations.** "#Param" indicates activated/total parameters.

| #Param | #Layers | #Heads | Hidden Size | Intermediate Size | #Experts | #Act. Experts |
|---|---|---|---|---|---|---|
| 0.3B/1.1B | 12 | 12 | 768 | 2048 | 16 | 2 |
| 0.8B/2.1B | 16 | 24 | 1536 | 1280 | 16 | 2 |
| 1.6B/7.3B | 16 | 16 | 2048 | 2048 | 32 | 4 |

Table 2: **Performance comparison across different training paradigms.**

| Method | ARC(e) | ARC(c) | PIQA | SciQ | OBQA | BoolQ | SIQA | WinoGrande | Avg. |
|---|---|---|---|---|---|---|---|---|---|
| Centralized | 49.7 | 24.4 | **68.9** | 74.0 | **30.6** | 54.3 | 42.0 | **53.5** | 49.7 |
| DiLiCo | **51.7** | **26.6** | 68.4 | 77.4 | 29.6 | 55.7 | **43.4** | 51.1 | 50.5 |
| SPES | **51.7** | 26.3 | 68.1 | **78.0** | 29.8 | **59.8** | 43.0 | 51.5 | **51.0** |

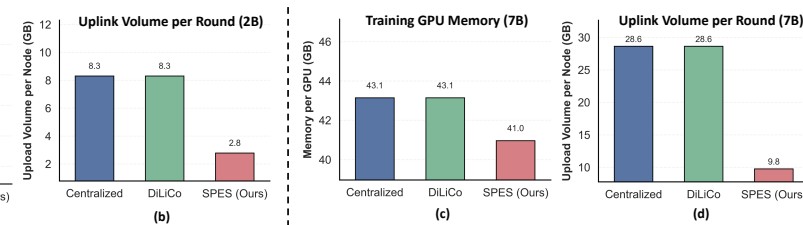

Figure 3: **Memory and communication costs for different training paradigms.** Experiments are conducted with a batch size of 2 and a sequence length of 2048. For the 2B parameter model, we employ standard PyTorch DDP. For the 7B parameter model, we utilize FSDP across 8 GPUs.

and robust multilingual support. For each node, the training data $\mathcal{D}_i$ for different nodes is randomly sampled from the whole dataset. Please refer to the **Appendix B** for more details.

**Evaluation Details.** We evaluate our model using the `lm-evaluation-harness` library (Gao et al., 2024) and report results on several commonsense reasoning benchmarks, including SIQA (Sap et al., 2019), ARC (easy and challenging) (Clark et al., 2018), SciQ (Johannes Welbl, 2017), PIQA (Bisk et al., 2020), OpenBookQA (Mihaylov et al., 2018), WinoGrande (Sakaguchi et al., 2021) and BoolQ (Clark et al., 2019). To assess general knowledge, we utilize MMLU (Hendrycks et al., 2020), CMMLU (Li et al., 2023), and C-Eval (Huang et al., 2023). Additional evaluation details are included in the **Appendix C**.

### 4.2 MAIN RESULTS

**Memory Cost Comparison.** Figs. 3 (a) and (c) compare the training memory footprints of SPES, DiLiCo, and centralized training. Both centralized training and DiLiCo require each node to update the full set of model parameters, resulting in high memory consumption. For example, training a 2B model requires more than 50GB memory per GPU, making it infeasible to train on commonly available 48GB GPUs. Furthermore, decentralized methods like DiLiCo cannot effectively leverage sharded training strategy due to limited inter-node bandwidth, further restricting the maximum trainable model size. In contrast, SPES keeps per-GPU memory under 40GB for a 2B model on 16 nodes without any sharding strategy. SPES can be combined with intra-node sharding for additional memory savings, as illustrated in Fig. 3(c). This efficiency arises from sparse training: each node updates only a subset of parameters, substantially reducing per-GPU memory.

**Communication Cost Comparison.** Figs. 3 (b) and (d) compare the communication overhead of different training schemes. In each round, both DiLiCo and centralized training need to upload the full set of model parameters, whereas SPES transmits only the updated parameters. In each communication round, both DiLiCo and centralized training require each node to upload the entire set of model parameters, whereas SPES only requires uploading the parameters that are actually updated. For instance, when training a 7B model on 4 nodes, SPES requires only 9.8GB data to be uploaded per node per round, compared to 28.6GB for DiLiCo and centralized training—a reduction of 65% in

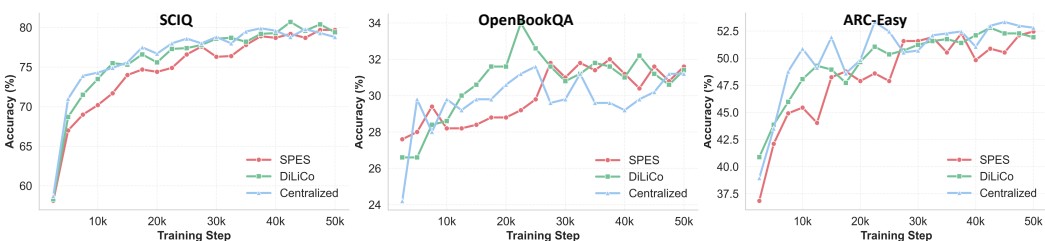

Figure 4: **Performance comparison across different training paradigms.** Performance during training is evaluated using the evaluation suite integrated into the open-source OLMo codebase.

Table 3: **Performance comparison with previous LLMs**.

| Method | #Params | #Tokens | SciQ | PIQA | SIQA | BoolQ | ARC(e) | ARC(c) |
|---|---|---|---|---|---|---|---|---|
| **Models Trained with Significantly More Tokens** | | | | | | | | |
| Qwen2.5-0.5B (Qwen et al., 2025) | 0.5B/0.5B | 18T | 93.0 | 69.9 | 47.1 | 61.7 | 64.6 | 35.8 |
| Qwen3-0.6B (Yang et al., 2025) | 0.6B/0.6B | 36T | 93.5 | 70.1 | 46.9 | 69.7 | 65.5 | 45.9 |
| Llama3.2-1B (Dubey et al., 2024) | 1.1B/1.1B | 9T | 91.3 | 73.7 | 45.0 | 63.7 | 71.6 | 43.5 |
| Qwen2.5-1.5B (Qwen et al., 2025) | 1.5B/1.5B | 18T | 94.1 | 75.8 | 53.5 | 72.6 | 75.3 | 53.9 |
| SmolLM2-1.7B (Allal et al., 2025) | 1.7B/1.7B | 11T | 93.2 | 77.4 | 46.7 | 72.4 | 77.8 | 54.1 |
| Qwen3-1.7B (Yang et al., 2025) | 1.7B/1.7B | 36T | 95.9 | 75.6 | 52.2 | 79.3 | 73.7 | 55.1 |
| OLMoE-1B-7B (Muennighoff et al., 2024) | 1.3B/7B | 5T | 94.9 | 80.6 | 47.8 | 74.4 | 78.0 | 55.2 |
| **Models with ≤ 3B Parameters** | | | | | | | | |
| OpenELM-0.5B (Mehta et al., 2024) | 0.5B/0.5B | 1.5T | 87.2 | 72.3 | - | 55.8 | 48.1 | 27.6 |
| MobiLlama-0.8B (Thawakar et al., 2024) | 0.8B/0.8B | 1.3T | 85.9 | 73.2 | 43.1 | 60.0 | 49.6 | 28.8 |
| TinyLlama-1.1B (Zhang et al., 2024b) | 1.1B/1.1B | 3T | 88.9 | 73.3 | - | 57.8 | 55.3 | 30.1 |
| OpenELM-1.1B (Mehta et al., 2024) | 1.1B/1.1B | 1.5T | 90.6 | 75.6 | - | 63.6 | 55.4 | 32.3 |
| OPT-1.3B (Zhang et al., 2022) | 1.3B/1.3B | 180B | 84.3 | 71.7 | 43.7 | 57.7 | 57.0 | 29.7 |
| MobiLlama-1.3B (Thawakar et al., 2024) | 1.3B/1.3B | 1.3T | 89.1 | 74.8 | 44.7 | 60.3 | 56.7 | 36.7 |
| Pythia-1.4B (Biderman et al., 2023) | 1.4B/1.4B | 300B | 86.4 | 70.9 | 44.6 | 63.3 | 60.7 | 31.2 |
| OPT-2.7B (Zhang et al., 2022) | 2.7B/2.7B | 180B | 85.8 | 73.1 | 44.1 | 60.4 | 60.8 | 34.0 |
| Pythia-2.8B (Biderman et al., 2023) | 2.8B/2.8B | 300B | 88.3 | 74.0 | 44.5 | 64.7 | 66.4 | 36.4 |
| Open-LLaMA-3B (Geng & Liu, 2023) | 3B/3B | 1T | 91.8 | 76.2 | - | - | 66.5 | 39.0 |
| **SPES-2B (ours)** | 0.8B/2.1B | 500B | 85.0 | 69.3 | 42.3 | 61.4 | 63.8 | 35.3 |
| **Models with ≥ 7B Parameters** | | | | | | | | |
| MoE++ 7B (Jin et al., 2024) | 1.2B/7B | 1T | 89.7 | 78.0 | 45.7 | 64.9 | 66.9 | 43.2 |
| LLaMA-MoE-3.0B (Zhu et al., 2024) | 3.0B/7B | 2.2T | 89.9 | 77.5 | - | - | 66.8 | 40.9 |
| OpenMoE-8B/32E (Xue et al., 2024) | 2.1B/8B | 1.1T | - | 74.2 | - | 61.2 | 64.1 | 30.3 |
| **SPES-7B (ours)** | 1.6B/7B | 500B | 89.9 | 74.7 | 44.8 | 62.7 | 72.1 | 43.8 |

uplink communication volume. This demonstrates the significant communication efficiency brought by the sparse training strategy of SPES.

**Training Speed Comparison.** We compare the training throughput of SPES against its centralized training counterpart. For the centralized setting, we adopt hybrid FSDP and train on four nodes, each equipped with 8×NVIDIA A800 GPUs and interconnected via RDMA. Each node contains four Mellanox InfiniBand HDR adapters, with each port operating at 100 Gbps (2×HDR lanes). In this configuration, centralized training reaches 3.79k tokens/s per GPU. Under the SPES setting (see the section of implementation details), throughput with $H = 50$ achieves 3.67k tokens/s. Despite running on a weaker hardware environment without high-bandwidth interconnects, SPES achieves a comparable speed. In addition, its throughput can be further improved by reducing the synchronization frequency, highlighting its scalability under resource-constrained conditions.

**Comparison with Previous Training Paradigms.** We evaluate SPES against both centralized training and the decentralized baseline DiLiCo, using 1B models trained on 100B tokens. As shown in Table 2, SPES achieves competitive performance on multiple benchmarks. Fig. 4 presents per-

Table 4: **Performance with and without expert merging.**

| Method | ARC(e) | ARC(c) | PIQA | SciQ | OBQA | BoolQ | SIQA | WinoGrande | Avg. |
|---|---|---|---|---|---|---|---|---|---|
| w/o merging | **52.8** | 26.5 | **68.4** | 75.9 | **30.0** | 58.0 | 42.4 | 50.3 | 50.5 |
| w/ merging | 52.1 | **27.7** | 67.4 | **77.8** | 28.8 | **60.4** | **42.7** | **53.5** | **51.3** |

formance trajectories during training. Although SPES exhibits a slightly slower initial learning curve, attributable to its sparse expert updates, it rapidly converges and ultimately matches or outperforms both baselines. Notably, SPES achieves this with substantially lower per-node GPU memory consumption and reduced synchronization bandwidth relative to centralized and decentralized alternatives. These results highlight that SPES provides a favorable trade-off between computational efficiency and model quality, enabling decentralized pretraining to attain competitiveness with large-scale centralized training under significantly lower resource budgets.

**Performance Comparison with Existing LLMs.** Finally, we compare our 2B and 7B models, which are trained with less than 500B tokens, with those open-source models of similar activation parameter scales and trained with less than 3T tokens. The results are shown in Table 3. We also show the results of models trained with significantly more tokens for reference.

We can see that across several commonsense reasoning benchmarks, both our 2B and 7B models consistently outperform most of their counterparts. It is worth noting that SPES-2B was trained in a decentralized manner on only 16 weakly connected 48GB GPUs, yet it remains competitive with models such as MobiLLama and OpenELM, which rely on substantially larger datasets and centralized infrastructures. This highlights the effectiveness of SPES in achieving strong performance under constrained hardware budgets. Moreover, SPES-7B attains results comparable to MoE++, which employs more advanced MoE designs (e.g., zero-computation experts) and larger training corpora. These findings indicate that SPES not only scales effectively and efficiently, but also retains significant room for improvement in architecture and data utilization, underscoring its potential as an extensible alternative to existing LLM training frameworks.

**Expert-Merging Warm-Up.** As shown in Table 4, utilizing expert merging increases the average score from 50.5 to 51.3, with notable improvements on BoolQ and SciQ. This indicates that cross-node parameter sharing enhances token utilization and promotes faster knowledge establishment, thus improving generalization across a range of reasoning and comprehension tasks.

*For ablation studies on key hyperparameters, including the merging factor $\alpha$, warm-up steps $T_{merge}$, local training steps $H$, and the number of nodes $N$, please refer to the **Appendix D** for details.*

## 5 CONCLUSION

We introduced SPES, a decentralized and memory-efficient pretraining paradigm for MoE-based LLMs. SPES assigned distinct subsets of experts to individual nodes and synchronized them only, substantially reducing per-device memory usage and communication overhead compared to centralized and prior decentralized approaches. To improve token utilization per expert, we introduced an expert-merging warm-up strategy to accelerate convergence in early training stages. Empirical results on 2B- and 7B-parameter MoE LLMs showed that SPES enabled efficient pretraining across weakly connected, geographically distributed GPU clusters, while achieving performance on par with comparable centralized baselines. Beyond lowering infrastructure demands, SPES broadened access to large-scale pretraining and could support more inclusive participation in LLM research, facilitating further advances in decentralized and memory-efficient training of foundation models.

**Limitations and Future Work.** Due to limited computational resources, our largest model comprises 7B parameters and was trained on a corpus of fewer than 500B tokens. Consequently, the scalability of our approach to significantly larger models or extended training contexts remains to be validated. A systematic exploration of these scaling behaviors represents an important direction for future research. In addition, our evaluation is confined to language understanding tasks in this work. In the future, we will investigate the applicability of SPES training to multimodal reasoning or generative tasks. Extending our framework to encompass these broader domains would provide a more comprehensive assessment of its generality and limitations.

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

APPENDIX

We provide the following materials in this appendix:

A. **Implementation Details**: more details of training hyper-parameters.

B. **Data Details**: dataset descriptions and sampling ratios.

C. **Evaluation Details**: evaluation datasets and metrics.

D. **Additional Results**: results on additional benchmarks and ablations on hyper-parameters.

E. **Declaration of LLM Assistance**: description of LLM usage in manuscript preparation.

## A. IMPLEMENTATION DETAILS

Table A1 shows our training configurations. For the 7B model, we train on the first 340B tokens using the settings specified in the table, then we reduce the per-node batch size to 1M tokens and set $H = 50$ to accelerate convergence. For the 2B model, we train on $440$B tokens under the default configuration, then reduce the per-node batch size to $0.5$M tokens and set $H = 50$.

For the 1B model, we perform ablation on expert-merging with a per-node batch size of 1024 to facilitate comparison with baselines trained under larger token budgets (400B). All other experiments use the hyper-parameters presented in Table A1. The training token budget is set to 100B for the ablations on $H$ and $N$, and 50B for $\alpha$ and $T_{merge}$ to allow faster validation. For all experiments, the loss coefficients are fixed across the models as follows: cross-entropy (1), load-balancing (0.01), MoE z-loss (0.001), and standard z-loss ($1 \times 10^{-5}$).

## B. DETAILS OF DATASETS AND SAMPLING RATIO

We train the model on data sampled from several open-source corpora, with sampling ratios provided in Table A2. Following OLMo et al. (2024), we apply a filter that removes all documents containing sequences of 32 or more repeated $n$-grams (an $n$-gram denotes any span of 1–13 tokens). The datasets used in our experiments are summarized as follows.

**Ultra-FineWeb.** Ultra-FineWeb (Wang et al., 2025) is a large-scale web corpus constructed from FineWeb (Penedo et al., 2024) and Chinese FineWeb (Yu et al., 2025) using an efficient verification-based filtering pipeline. The approach combines lightweight fastText classification with a verification mechanism, enabling reliable data selection at substantially reduced computational cost. The final corpus comprises roughly 1 trillion English tokens and 120 billion Chinese tokens. By enhancing overall data quality, Ultra-FineWeb provides a strong foundation for LLM training and contributes to the dataset used in MiniCPM4 (Team et al., 2025).

**SlimPajama.** SlimPajama (Soboleva et al., 2023) is a large-scale, rigorously deduplicated corpus constructed from RedPajama (Weber et al., 2024). Using a multi-stage pipeline that combines qual-

Table A1: **Training hyperparameters for different model scales.**

|  | 7B | 2B | 1B |
|---|---|---|---|
| Maximum Learning Rate | $4 \times 10^{-4}$ | $5 \times 10^{-4}$ | $5 \times 10^{-4}$ |
| Minimum Learning Rate | $4 \times 10^{-5}$ | $5 \times 10^{-5}$ | $5 \times 10^{-5}$ |
| Optimizer $\epsilon$ | $1 \times 10^{-8}$ | $1 \times 10^{-8}$ | $1 \times 10^{-8}$ |
| Weight Decay | 0.1 | 0.1 | 0.1 |
| $(\beta_0, \beta_1)$ | (0.9, 0.95) | (0.9, 0.95) | (0.9, 0.95) |
| LR Warmup Steps | 2000 | 2000 | 2000 |
| Sequence Length | 2048 | 2048 | 2048 |
| Batch Size (Tokens) | 1M $\times 16$ | 2M $\times 4$ | 0.5M $\times 4$ |
| Synchronization Steps $H$ | 100 | 100 | 50 |

Table A2: **Sample ratios of different datasets.**

| Dataset | Ultra-FineWeb | SlimPjama | StarCoder | arXiv | OpenWebMath | Pes2o | Algebric Stack |
|---|---|---|---|---|---|---|---|
| **Ratio (%)** | 64.2 | 27.2 | 6.6 | 0.7 | 0.4 | 0.5 | 0.4 |

ity filtering with MinHashLSH-based deduplication at trillion-token scale, SlimPajama substantially reduces redundancy and low-quality content, compressing the dataset from 1.21T to 627B tokens while retaining domain coverage. The corpus spans diverse sources, including CommonCrawl, C4, GitHub, Books, ArXiv, Wikipedia, and StackExchange.

**OLMo-Mix-1124.** OLMo-Mix-1124 is a 3.9-trillion-token corpus comprising over 95% web data, constructed from DCLM (Li et al., 2024), Dolma v1.7 (Soldaini et al., 2024), and Star-Coder (Lozhkov et al., 2024). For our work, we extract scientific-domain subsets, including arXiv, OpenWebMath, Algebraic Stack, peS2o, and StarCoder.

## C. EVALUATION DETAILS

We evaluate our models with the `lm-evaluation-harness` library (Gao et al., 2024), which offers standardized benchmark implementations and facilitates direct comparison with prior work. All experiments use version 0.4.7. The benchmarks and evaluation settings are detailed below:

**SciQ** (Johannes Welbl, 2017) is a science multiple-choice question-answering dataset. The questions were generated by crowdworkers and validated against science reference materials, covering topics such as physics, biology, and chemistry. As the questions are designed to resemble real exam-style queries, the dataset tests scientific knowledge and reasoning skills of a model. We report 0-shot accuracy on SciQ.

**ARC** (Clark et al., 2018) (AI2 Reasoning Challenge) consists of grade-school level science exam questions, partitioned into ARC-Easy (ARC-E) and ARC-Challenge (ARC-C). ARC-E contains questions that can often be answered by retrieval of surface-level facts, while ARC-C includes the more demanding questions requiring reasoning and multi-step inference across scientific facts. We report 0-shot accuracy on ARC-E and 25-shot normalized accuracy on ARC-C.

**SIQA** (Sap et al., 2019) (SocialIQA) benchmarks social commonsense reasoning. Each instance presents a short human-centered scenario alongside a question about likely intents, causes, or outcomes of human actions. This evaluates the model's ability to handle subtle social reasoning and cause-effect relationships in naturalistic settings. We report 0-shot normalized accuracy on SIQA.

**PIQA** (Bisk et al., 2020) (Physical Interaction QA) evaluates physical commonsense reasoning in everyday situations. Given a description of a goal, the model must choose the most plausible solution among two alternatives, testing physical feasibility and everyday world knowledge. We report 0-shot normalized accuracy on PIQA.

**OpenBookQA** (Mihaylov et al., 2018) presents multiple-choice science questions paired with a small open-book of 1,326 core scientific facts. Answering the questions typically requires combining knowledge from the book with additional commonsense reasoning, making this benchmark particularly challenging. We report 0-shot normalized accuracy on OpenBookQA.

**WinoGrande** (Sakaguchi et al., 2021) is a large-scale dataset for pronoun resolution, created to reduce annotation artifacts common in earlier benchmarks (e.g., Winograd Schema Challenge). Each instance requires the model to resolve ambiguous pronouns based on contextual clues, testing commonsense reasoning and language understanding. We report 0-shot accuracy on WinoGrande.

**BoolQ** (Clark et al., 2019) is a reading comprehension dataset in the yes/no QA format. Questions are naturally occurring user queries, paired with passages from Wikipedia that may or may not contain the answer. Models must perform passage-level understanding to correctly infer the response. We report 0-shot accuracy on BoolQ.

**MMLU** (Hendrycks et al., 2020) (Massive Multitask Language Understanding) covers 57 tasks across diverse domains such as mathematics, history, law, medicine, and the natural sciences. As

Table A3: **Performance comparison with previous LLMs on additional benchmarks.** Some models are excluded because they neither report results on these benchmarks nor are compatible with `lm-evaluation-harness`.

| Method | #Params | #Tokens | OBQA | MMLU | CMMLU | C-Eval |
|---|---|---|---|---|---|---|
| **Models Trained with Significantly More Tokens** | | | | | | |
| Qwen2.5-0.5B (Qwen et al., 2025) | 0.5B/0.5B | 18T | 35.4 | 47.3 | 49.5 | 51.0 |
| Qwen3-0.6B (Yang et al., 2025) | 0.6B/0.6B | 36T | 34.2 | 52.8 | 50.4 | - |
| Llama3.2-1B (Dubey et al., 2024) | 1.1B/1.1B | 9T | 36.2 | 36.6 | 29.4 | 30.9 |
| Qwen2.5-1.5B (Qwen et al., 2025) | 1.5B/1.5B | 18T | 40.4 | 59.7 | 66.3 | 68.2 |
| SmolLM2-1.7B (Allal et al., 2025) | 1.7B/1.7B | 11T | 43.6 | 48.4 | 31.0 | 32.5 |
| Qwen3-1.7B (Yang et al., 2025) | 1.7B/1.7B | 36T | 38.6 | 62.6 | 68.1 | - |
| OLMoE-1B-7B (Muennighoff et al., 2024) | 1.3B/7B | 5T | 45.2 | 50.5 | 31.9 | 31.1 |
| **Models with $\leq 3B$ Parameters** | | | | | | |
| MobiLlama-0.8B (Thawakar et al., 2024) | 0.8B/0.8B | 1.3T | 33.0 | 23.5 | 25.3 | 22.7 |
| TinyLlama-1.1B (Zhang et al., 2024b) | 1.1B/1.1B | 3T | 36.8 | 25.3 | 24.9 | 26.0 |
| OPT-1.3B (Zhang et al., 2022) | 1.3B/1.3B | 180B | 33.4 | 24.9 | 25.3 | 23.0 |
| MobiLlama-1.3B (Thawakar et al., 2024) | 1.3B/1.3B | 1.3T | 35.4 | 25.3 | 23.5 | 26.2 |
| Pythia-1.4B (Biderman et al., 2023) | 1.4B/1.4B | 300B | 33.4 | 24.2 | 25.6 | 23.0 |
| OPT-2.7B (Zhang et al., 2022) | 2.7B/2.7B | 180B | 35.2 | 25.6 | 25.3 | 23.0 |
| Pythia-2.8B (Biderman et al., 2023) | 2.8B/2.8B | 300B | 35.6 | 25.2 | 25.4 | 22.9 |
| **SPES-2B (ours)** | 0.8B/2.1B | 500B | 31.4 | 25.5 | 24.9 | 25.0 |
| **Models with 7B Parameters** | | | | | | |
| MoE++ 7B (Jin et al., 2024) | 1.2B/7B | 1T | 40.0 | 25.1 | 24.9 | 23.6 |
| **SPES-7B (ours)** | 1.6B/7B | 500B | 39.4 | 24.1 | 25.0 | 26.2 |

a broad knowledge benchmark, it measures both factual recall and domain-specific reasoning. We follow standard settings and report 0-shot accuracy on MMLU.

**CMMLU** (Li et al., 2023) is the Chinese adaptation of MMLU. It mirrors the structure of MMLU but uses Chinese linguistic and cultural contexts, making it suitable for evaluating reasoning and domain knowledge in the Chinese language. We report 0-shot accuracy on C-MMLU.

**C-Eval** (Huang et al., 2023) is a comprehensive Chinese evaluation suite consisting of over 13,000 multiple-choice questions spanning 52 subjects, from elementary school topics to professional certification exams. It provides a fine-grained view of model performance in academic and professional domains under Chinese cultural and linguistic settings. We report 0-shot accuracy on C-Eval.

## D. ADDITIONAL RESULTS

**Results on Additional Benchmarks.** Table A3 reports the performance of our models on additional benchmarks. On Chinese evaluation datasets, SPES-7B surpasses the comparable baseline MoE++ (26.2 vs. 23.6 on C-Eval; 25.0 vs. 24.9 on CMMLU), while maintaining competitive performance on other tasks. This indicates that SPES can match the performance of centrally trained models under resource-constrained settings, underscoring its potential to lower the barrier to LLM pretraining. In addition, SPES-2B attains performance on par with models of similar scale using only 16 weakly connected nodes, further validating the efficiency of our approach.

**Ablation on Hyperparameters in Expert Merging.** Fig. A1 shows the effect of varying merging warmup steps $T_{merge}$ and the merging factor $\alpha$ on performance. A moderate warmup of 12.5k steps achieves the best results, as shorter schedules hinder sufficient knowledge exchange, while excessively long ones interfere with expert specialization. Similarly, performance peaks when $\alpha$ is set to 0.1, with both smaller and larger values leading to degradation. These observations suggest that effective expert merging requires a careful balance between inter-expert knowledge sharing and expert specialization. Overly aggressive merging may overwrite expert-specific information, whereas

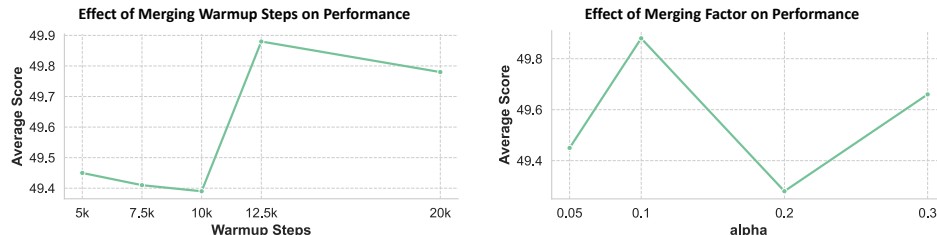

Figure A1: **Ablation on key hyper-parameters in expert merging.** The reported average is computed over ARC(e), SciQ, PIQA, WinoGrande, ARC(c), OBQA, OpenBookQA, and SIQA.

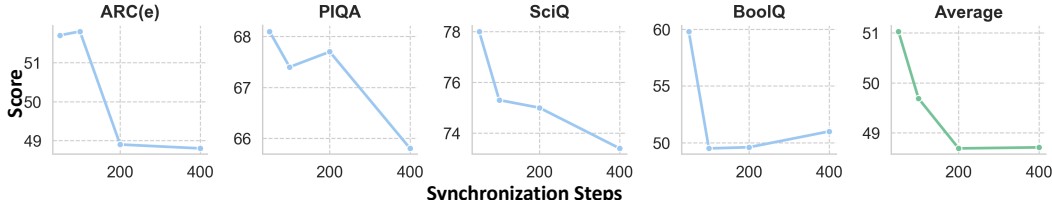

Figure A2: **Ablation on synchronization steps.** The reported average is computed over eight benchmarks in total, additionally including ARC(c), OBQA, OpenBookQA, and SIQA.

Table A4: **Performance comparison with different numbers of nodes.**

| No. of Nodes | ARC(e) | ARC(c) | PIQA | SciQ | OBQA | BoolQ | SIQA | WinoGrande | Avg. |
|---|---|---|---|---|---|---|---|---|---|
| 2 | 52.0 | 25.7 | 68.7 | 77.6 | 30.4 | 58.0 | 42.2 | 50.4 | 50.6 |
| 4 | 51.8 | 27.4 | 67.4 | 75.3 | 29.8 | 49.5 | 43.7 | 52.6 | 49.7 |
| 8 | 47.9 | 24.6 | 66.3 | 70.8 | 29.4 | 60.1 | 42.8 | 53.9 | 49.5 |

insufficient merging yields only minor parameter updates and limits the efficiency of knowledge sharing across experts, thereby slowing the establishment of general expert representations.

**Ablation on Number of Nodes.** We then study the impact of varying the number of nodes $N$ while keeping the global batch size fixed. As shown in Table A4, model performance remains stable when scaling from 2 to 8 nodes. The average score decreases slightly from 50.6 (2 nodes) to 49.5 (8 nodes), yet SPES maintains competitive results across benchmarks. This behavior illustrates a natural trade-off in decentralized sparse training: increasing the number of nodes leads to greater fragmentation of training data and experts, which can modestly slow convergence. Nonetheless, the results underscore the robustness of SPES. Even with reduced per-node token utilization, it maintains overall performance. These findings demonstrate SPES' potential of scalability, suggesting that it can effectively leverage a larger number of participants while maintaining model quality, a key property for practical deployment in heterogeneous, distributed environments.

**Ablation on Synchronization Steps.** We analyze the effect of varying the local update interval $H$ in the SPES framework. As illustrated in Fig. A2, performance declines when $H$ increases from 50 to 200 or 400. This trend reflects a key trade-off in decentralized sparse training: while larger $H$ reduces communication frequency, it amplifies model divergence across nodes, weakening the benefits of expert sharing. Overall, $H = 50$ provides the best balance between communication efficiency and model quality, underscoring the necessity of frequent synchronization to fully exploit SPES' sparse expert updates under bandwidth-limited decentralized settings.

## E. DECLARATION OF LLM ASSISTANCE

We use ChatGPT-5 to assist with the refinement of this manuscript. After drafting the full text, we provided selected passages to the models for suggestions on grammar, clarity, and conciseness. All revisions were reviewed and finalized by the authors to ensure accuracy and appropriateness.

