# OpenReview forum: "Pretraining a Large Language Model using Distributed GPUs: A Memory-Efficient Decentralized Paradigm"
_ICLR.cc/2026/Conference — ICLR 2026 Conference Withdrawn Submission_

### Official Review · Reviewer_pbzh · 2025-10-27

**Soundness:** 2
**Presentation:** 3
**Contribution:** 2
**Rating:** 4
**Confidence:** 4

**Summary:**

The paper proposes a decentralized pre-training paradigm for Mixture-of-Experts (MoE) LLMs. Each node trains only a subset of experts, reducing optimizer state and gradient memory. Periodic sparse synchronization updates shared parameters and only the trained experts, lowering communication volume. To address reduced token utilization per expert, an expert-merging warm-up aligns similar experts early in training to accelerate convergence.
Experiments include 1B, 2B, and 7B parameter MoE models trained on public datasets over low-bandwidth Ethernet. The 16-node, single-GPU-per-node 2B training demonstrates up to 33% communication savings and reduced GPU memory from ~55 GB to ~35 GB per node. Results show competitive downstream accuracy with centralized and DiLiCo baselines using comparable compute. The paper shows small-scale scalability to 7B with 4 nodes, each containing 8 GPUs.

**Strengths:**

- The focus on low-memory GPUs and commodity networking broadens LLM pretraining access beyond industrial clusters.
- Decentralized expert sharding is a well-motivated extension of MoE structures for memory-constrained training.
- Communication and memory accounting is explicit and favorable vs. DiLiCo and centralized baselines.

**Weaknesses:**

- Comparisons are limited to older or non-SOTA MoE baselines. Strong contemporary baselines like DeepSeek-MoE and Mixtral-style routing are not benchmarked.
- End-task performance remains modest. For example, SPES-2B underperforms several 2–3B centralized models trained on similar or fewer tokens (Table 3).
- Sparse convergence behavior lacks theoretical support. Practical convergence seems slower early on, but the paper does not deeply analyze variance effects, expert imbalance, or routing drift.
- Some architectural and hyperparameter choices are under-justified (e.g., K in Top-K merging, warm-up length, decay schedule).
- Fixed expert assignment may cause uneven specialization and compute utilization; adaptive allocation or load-aware routing could be relevant.
- Evaluation domains are narrow. Only language understanding tasks are assessed, although MoE benefits could be stronger on multilingual/generative reasoning.

**Questions:**

- How does SPES handle node dropout or stragglers during synchronization?
- Could expert partitioning be made dynamic, adjusting to load, routing probabilities, or token distributions?
- What is the impact of increasing synchronization period H beyond reported settings on stability and final accuracy?
- How sensitive is merging effectiveness to expert similarity metric choice?
- Does expert merging produce mode collapse, reducing specialization later in training?

---

### Official Review · Reviewer_5cjC · 2025-10-28

**Soundness:** 2
**Presentation:** 2
**Contribution:** 2
**Rating:** 2
**Confidence:** 4

**Summary:**

1. The paper proposes SParse Expert Synchronization (SPES), a decentralized MoE pretraining framework where each node trains a distinct, locally assigned subset of experts. This heterogeneous model assignment is enabled by infrequent, federated-style synchronization.
2. This sparse assignment significantly reduces memory consumption, as gradients and optimizer states are stored only for local experts, and cuts communication overhead by transmitting only updated parameters. SPES achieves comparable training throughput and competitive performance with no degradation compared to baselines.
3. To facilitate knowledge sharing, an expert-merging warm-up strategy is introduced. This method periodically averages parameters of similar experts, which are identified by computing the cosine similarity of their input projection layers.

Despite achieving strong empirical results, the work is constrained by limited methodological novelty, a lack of analytical rigour, and an overly complex training pipeline.

**Strengths:**

1. The parameter-partitioning method is highly effective at reducing memory and communication costs.
2. The proposed expert warmup seems to work empirically, despite a lack of theoretical justification.

**Weaknesses:**

1.  Limited Methodological Novelty and Suboptimal Memory Savings: The core concept of training replica-specific parameters is a well-established technique in fields such as Federated Learning[1] and communication-efficient pre-training[2]. The paper's contribution is thus an application of this known paradigm to the MoE layers, which does not in itself constitute a significant novel contribution. Furthermore, the implementation's benefits are constrained; unlike prior work that fully partitioned models to save on parameter memory, the proposed method requires each node to store a full copy of all model parameters. The memory savings are thus limited to optimizer states and gradients.
2.  Lack of Theoretical Rigour: The paper is presented as an empirical report, lacking any formal theoretical analysis of the proposed method. This positions the work as a systems-level contribution.
3.  Ad-Hoc Warmup Strategy and Unmeasured Expert Redundancy: The proposed expert-merging warmup is an ad-hoc heuristic that actively encourages parameter similarity, potentially leading to expert redundancy. While the authors posit that limiting the warmup's duration mitigates this risk, this claim is unsubstantiated. The paper provides no direct, quantitative analysis of expert similarity, routing distributions, or specialization metrics post-warmup. The provided ablations on warmup duration only report downstream task performance, failing to explicitly measure the strategy's impact on expert redundancy.
4.  Unjustified Similarity Metric: The strategy relies on cosine similarity of input projection layers to identify candidates for merging. The assumption that this linear metric is a valid proxy for beneficial gradient sharing across multiple nodes within a non-linear model is presented without analytical proof. This heuristic lacks a formal justification, and it cannot be trivially equated to gradient aggregation or effective batch size increases.
5.  Omission of Simpler Baselines: The paper fails to justify the necessity of its complex expert-merging strategy over simpler, established alternatives. A standard Data-Parallel (DDP) warmup phase, for example, would provide a stable and well-understood initialization with theoretical guarantees for all experts while incurring a cost that is negligible when amortized over the full pre-training run. The paper does not benchmark against this conventional warmup strategy, leaving the added complexity of the proposed method entirely unmotivated.

[1] Arivazhagan, et.al; "Federated Learning with Personalization Layers"

[2] Iacob, et.al; "DEPT: Decoupled Embeddings for Pre-training Language Models"

**Questions:**

1. Have you benchmarked the use of a warmup with standard DDP to initialize the expert layers before partitioning? While I understand that memory costs may be prohibitive, such warmups are usually quite short and could tolerate memory-saving techniques like activation checkpointing, CPU-offloading, and so on.
2. Could you please provide direct measurements of expert similarity for various warmup durations?

---

### Official Review · Reviewer_xjtR · 2025-10-30

**Soundness:** 3
**Presentation:** 3
**Contribution:** 3
**Rating:** 4
**Confidence:** 4

**Summary:**

The paper proposes SPES (SParse Expert Synchronization), a decentralized pretraining framework for MoE LLMs where each node trains only a subset of experts and periodically synchronizes those experts, not the full model. It also introduces an expert-merging warm-up to accelerate convergence. Experiments report training a 2B MoE on 16×48 GB L40S over the public internet with competitive quality and lower memory/communication than prior decentralized methods, and a 7B MoE that matches centralized baselines.

**Strengths:**

The paper introduces a practical and original approach to decentralized large language model pretraining through sparse expert synchronization (SPES), where each node trains only a subset of experts and exchanges parameters selectively. This design substantially reduces memory and communication costs, making large-scale training feasible on commodity GPUs. The addition of the expert-merging warm-up is an inventive technique that improves convergence stability and reduces early-stage inefficiencies. The proposed framework is empirically well-validated, with convincing 2B and 7B experiments showing competitive results against centralized and existing decentralized baselines. The writing is clear, the figures are informative, and the work addresses an important problem of accessibility and scalability in modern LLM training.

**Weaknesses:**

- Limited detail on routing load balance and expert under-utilization in heterogeneous  (non-IID) nodes.
- A key innovation of SPES is the expert-merging warm-up strategy, where similar experts across nodes are merged early in training to accelerate convergence. However, the paper doesn’t delve deeply into how these merges affect optimization dynamics or whether repeated merging can destabilize learning.  How robust is that merging criterion to noise early in training? Without theoretical justification or ablation studies comparing different merge criteria, it’s hard to assess how sensitive the approach is or whether it generalizes to other setups.
- Miss reference regarding decentralized LLM pre-training: [1] Douillard, Arthur, et al. "Dipaco: Distributed path composition." arXiv preprint arXiv:2403.10616 (2024). [2] Iacob, Alex, et al. "DES-LOC: Desynced Low Communication Adaptive Optimizers for Training Foundation Models." arXiv preprint arXiv:2505.22549 (2025).

**Questions:**

- How sensitive is performance to sync period, expert assignment, and merge frequency/weights?
- How does the expert-merging warm-up affect model converge?
- What mechanisms prevent expert collapse or drift after repeated merges?
- Can SPES handle non-IID data across nodes without harming specialization?
- Please report token throughput vs. bandwidth and wall-clock vs. DiLiCo/Photon on the same hardware.

---

### Official Review · Reviewer_VHpS · 2025-10-31

**Soundness:** 2
**Presentation:** 2
**Contribution:** 2
**Rating:** 2
**Confidence:** 5

**Summary:**

This work tackles an interesting and important problem of optimizing the communication and computation efficiency of decentralized pre-training of MoE-based LLMs. The authors propose a sparse expert synchronization (SPES) method that leverages the structural properties of MoE architectures to train small subsets across decentralized computing silos. SPES combines local updates with an expert-merging warmup phase to accelerate convergence and enable efficient knowledge sharing. The authors used SPES to train across 16 decentralized silos with limited hardware accelerators and a 2B-parameter model. Using SPES, the authors achieve competitive performance compared to centrally trained LLMs under similar budgets, and demonstrate scalability up to a 7B-parameter model.

**Strengths:**

- S1. This work studies a timely and critical problem to improve upon efficient decentralized pre-training techniques: reducing the memory footprint and computational overhead, and further reducing the communicated bytes.
- S2. The SPES method provides a clear reduction in communication costs and per-GPU memory requirements, allowing for a more flexible decentralized setting and loosening the requirements for participating in collaborative pre-training.
- S3. The paper creatively incorporates a model-merging strategy during warm-up, drawing on techniques from the model-merging literature to regularize knowledge sharing between experts.

**Weaknesses:**

- W1. SPES still requires each node to hold a full model copy (all experts), meaning that while each node only trains its subset, the entire model parameters must reside and eventually be synced on every node. This limits memory savings and does not reduce the downstream communication from the server to workers during synchronization.
- W2. The effect of Top-K routing in the SPES method on FLOPs is not sufficiently discussed, which limits the clarity of this work’s contributions.
- W3. The use of load balancing losses is not sufficiently discussed in the context of SPES, where the number of trainable local experts is not the same as the number of total trainable experts.
- W4. The model merging technique applied during the warm-up phase lacks a clear theoretical justification. Previous literature in the area focused on using routing network weights or down-projection weights as proxies for experts' similarity rather than up-projections.
- W5. A systematic study of the scalability of the SPES approach is missing. The community would benefit significantly from understanding the interplay among scalable factors such as the number of decentralized workers, model size, and network bandwidth.
- W6. Experiments and comparisons are conducted with unusual global batch sizes, which massively reduce gradient noise (thus, hiding potential convergence problems) at the cost of a lot of compute (which scales linearly with the batch size).

**Questions:**

- Q1. Can the authors discuss more about the Top-K element in the routing network and its impact on the inactive local experts in section 3.3?
- Q2. Can the authors explain why they used FedAvg as the outer optimizer instead of DiLoCo?
- Q3. Can the authors justify their choice of the warm-up phase expert merging more extensively? For example, why do the authors use the up-projection to assess expert similarity? In addition, according to the ablation studies in the appendix, it is quite unclear what the impact of the merging phase is, considering that the performance doesn’t change significantly when varying the hyperparameters involved. Can the authors also clarify this aspect?
- Q4. Can the authors extensively explain how the load balancing losses are implemented in the context of local inactive experts and what effect they expect them to have?
- Q5. Can the authors provide a more extensive evaluation with respect to the number of decentralized workers?
- Q6. Can the authors clarify lines 366-368? Making quantitative examples would help significantly.
- Q7. Can the authors justify the fact that the initial values of the performance in Figure 4 differ across methods for OpenBookQA and ARC-Easy but are the same for SCIQ? Is this expected? What may cause this discrepancy?
- Q8. Most analyses in section 4.2 are too qualitative and could be more insightful. For example, in the “Training Speed Comparison”, can the authors clarify why they obtain a similar throughput despite communicating 50 times less frequently over a network about 10 times slower?
- Q9. I found the table A1 very concerning, in particular, the line explaining the batch size used. Can the authors extensively explain what their choice of the batch size (local and global) comes from? How did they execute the tuning? How did they count the number of tokens processed?

---

### Note · Authors · 2025-11-12

I have read and agree with the venue's withdrawal policy on behalf of myself and my co-authors.